# Microglia and Cholesterol Handling: Implications for Alzheimer’s Disease

**DOI:** 10.3390/biomedicines10123105

**Published:** 2022-12-01

**Authors:** Oscar M. Muñoz Herrera, Angela M. Zivkovic

**Affiliations:** Department of Nutrition, University of California, Davis, CA 95616, USA

**Keywords:** microglia, cholesterol, Alzheimer’s disease

## Abstract

Cholesterol is essential for brain function and structure, however altered cholesterol metabolism and transport are hallmarks of multiple neurodegenerative conditions, including Alzheimer’s disease (AD). The well-established link between apolipoprotein E (APOE) genotype and increased AD risk highlights the importance of cholesterol and lipid transport in AD etiology. Whereas more is known about the regulation and dysregulation of cholesterol metabolism and transport in neurons and astrocytes, less is known about how microglia, the immune cells of the brain, handle cholesterol, and the subsequent implications for the ability of microglia to perform their essential functions. Evidence is emerging that a high-cholesterol environment, particularly in the context of defects in the ability to transport cholesterol (e.g., expression of the high-risk APOE4 isoform), can lead to chronic activation, increased inflammatory signaling, and reduced phagocytic capacity, which have been associated with AD pathology. In this narrative review we describe how cholesterol regulates microglia phenotype and function, and discuss what is known about the effects of statins on microglia, as well as highlighting areas of future research to advance knowledge that can lead to the development of novel therapies for the prevention and treatment of AD.

## 1. Introduction

Since Alzheimer’s disease (AD) was first described by Alois Alzheimer over a century ago [1], it has been known that an aberrant accumulation of lipid “saccules” as they were called then, essentially an enrichment in intracellular lipids, is characteristic of the brain in AD. Since then we have gained a much more sophisticated understanding of how lipid accumulation and aberrant lipid metabolism contribute to the etiology of AD. It is now clear, for example, that cholesterol enrichment in the plasma membranes of neurons is causally linked to production of the cytotoxic amyloid beta (Aβ) peptide [2] through a lipid raft-dependent mechanism. Cholesterol is an essential lipid component for various cellular structures and organelles, and its trafficking occurs via various distinct pathways which include endocytosis/phagocytosis, transport to the plasma membrane and repurposing of cholesterol by removal from the plasma membrane [3,4,5,6,7,8,9,10,11,12,13,14,15,16,17]. Disruption in cholesterol transport and trafficking can lead to aberrant cellular function. For example, in Niemann-Pick Type C disease, which involves an autosomal recessive mutation causing neurodegeneration, aberrant cholesterol metabolism is linked to Aβ deposition similar to the pathology occurring in AD [18]. In mouse models hypercholesterolemia results in glial cell hyperactivation, accelerating amyloid pathology in the brain [19]. In zebrafish exposure to high cholesterol (4% weight/weight cholesterol) for 19 days resulted in higher brain mRNA expression of proinflammatory markers and elevated brain mRNA of cluster of differentiation molecule 11B, a microglia marker, in a type 2 diabetes model [20].

However, it is not yet clear whether unregulated cholesterol drives pathology or if cholesterol is simply worsening an already pathological process. Much more is known about the cholesterol-mediated regulation of cellular function in neurons and astrocytes in the context of AD [21,22,23]; however not as much is known about how microglia, the immune cells of the brain, respond to high cholesterol environments. Microglia are quickly emerging as key players in the pathophysiology of AD [24,25,26,27,28,29]. Genome-wide association studies have consistently found that genes expressed predominantly or exclusively in microglia in the brain, such as triggering receptor expressed on myeloid cells 2 and myeloid cell surface antigen CD33, are associated with AD risk [30,31,32,33,34]. Disease-associated microglia (DAMs), or activated microglia found to be enriched in plaque areas of the AD brain [25,35,36,37,38,39], are directly linked to neuroinflammation, which has been shown to be a critical driver of pathology in AD [40]. Although a vast literature from the cardiovascular field points to cholesterol accumulation in macrophages, the peripheral cousins of microglia, in causing the formation of “foam cells” which are in turn causally linked to the process of atherosclerosis [41,42,43], not much is known about how microglia respond to excessive cholesterol accumulation.

Multi-omic analysis of postmortem brain samples from AD patients revealed that the concentrations of free cholesterol were elevated in microglial endocytic vesicles [44] suggesting that microglial cholesterol handling may be directly involved in AD pathology. Importantly, it has been shown that the ability of microglia to degrade Aβ is dependent on their ability to efflux cholesterol [45]. Thus, there is already compelling evidence that cholesterol-mediated regulation of microglial function may be an important contributing factor in AD. Here, we review what is known about cholesterol-mediated regulation of microglia phenotype and function, discuss the impacts of statins, which inhibit the synthesis of cholesterol, on microglia function, and highlight areas of future research in the context of AD.

## 2. Cholesterol-Mediated Regulation of Microglia Phenotype

Microglia are classically characterized as immune cells predominantly found in a homeostatic state, only moving outside of that resting or homeostatic phenotype to respond acutely to the infiltration of foreign and harmful substances [46,47]. However, recent studies have revealed that microglia are in fact very dynamic, occupying a diverse set of states, with different functional phenotypes [48,49]. Microglia perform different roles in different environments, falling into four broad categories: injury-response microglia, proliferative-region-associated microglia (PAM), disease-associated microglia (DAM), and lipid droplet-accumulating microglia. Injury-response microglia can be induced by lysolecithin injection in mice [50], which leads to the upregulation of lipoprotein lipase and apolipoprotein E (APOE) among other effects [50]. Microglia that surround oligodendrocytes during the first week after birth are highly phagocytic, identified as PAM, and share a very similar profile at the transcriptional level with DAM, such as the increased expression of lipid metabolic genes [51]. Researchers have identified the transition into the DAM state by the downregulation of typical microglia markers (e.g., C-X3-C motif chemokine receptor 1 and adenosine diphosphate receptor P2Y12) and the activation of phagocytic and lipid metabolic genes; as well as having overlapping signatures with injury-response microglia and PAM [50,51]. Elements of the DAM profile are highly conserved from PAM to DAM in mouse models [36,51], behaving with similar phenotypic states as PAM in the zebrafish model [52], indicating that both processes are highly regulated. Even though the murine DAM gene profile from an AD-inducible model is detected in human DAM, the majority of those genes are dispersed across multiple subgroups of microglia instead of being specific to DAM in humans, highlighting species-specific differences in microglia phenotype [53]. 

Lipids are essential to the brain for both structure and function [54]; however, an overload of lipids can lead to catastrophic damage. When lipid molecules in a given area surpass a critical concentration they begin to aggregate, form micelles, and even form crystals (e.g., cholesterol crystals), which then act as detergents and structures that lyse and otherwise damage cells and their components [55]. Thus, lipid storage and disposal mechanisms have emerged to protect cells from these extreme events. Cells usually respond to excess cholesterol by inducing the cholesterol efflux machinery, including APOE [56]. Although beyond the scope of this review article, cholesterol metabolism and regulation have been described thoroughly elsewhere [57]. Briefly, the sterol regulatory element binding protein family is involved in regulating genes involved in cholesterol synthesis, transport, and efflux. Induction of the sterol regulatory element binding protein transcriptional program in response to low or high concentrations of cholesterol detected in the endoplasmic reticulum membrane activates and deactivates, respectively, the transcription of genes involved in lipid biosynthesis and import vs. storage and efflux [58,59], including the expression of low-density lipoprotein receptor [58,60] for cholesterol import, and the adenosine triphosphate-binding cassette (ABC) family, in particular ABCA1, for cholesterol efflux [61,62,63]. However, when the capacity to utilize or efflux excess lipids has been surpassed all cells have the ability to form lipid droplets as a way to temporarily store the excess [64]. When the storage of lipids in lipid droplets becomes chronic and/or the number of lipid droplets starts to exceed the normal threshold, there can be effects on the ability of cells to perform their normal functions [65]. 

In the case of microglia, lipid droplet-accumulating microglia, or microglia that are characterized by lipid droplet accumulation, perform with a defective phagocytic phenotype [66]. Lipid droplet-accumulating microglia have an enhanced phagocytic uptake of lipid, exacerbating the lipid droplet accumulation burden, and promoting chronic and self-sustained microglial activation [67]. Sustained inflammation further pushes microglia into an activated state of hyperactivity, which creates a feedback that exacerbates neuroinflammation and damages blood–brain barrier integrity [68]. Whereas cholesterol excess leads to lipid droplet accumulation and chronic microglial activation, dysregulated cholesterol concentration in the opposite direction can also be problematic. For example, in mouse models with an interleukin-10 receptor knockout (specifically in astrocytes) there is prolonged neuroinflammatory response to peripheral lipopolysaccharide (LPS), with interleukin-10 receptor signaling deficits and a lack of cholesterol biosynthesis both leading to the inability to resolve microglial activation [69]. It is still unclear in which context injury-response microglia, PAM and DAM become aggressors vs. protectors [70], but there is ongoing research to understand the dynamics of change in microglia phenotype so that we can better understand how to intervene to modulate phenotype to prevent or decrease neurodegeneration [71].

## 3. Cholesterol-Mediated Regulation of Microglia Function

One of the main functions of microglia is to remove debris and other cytotoxic molecules in a constant effort to maintain a homeostatic environment [48,72]. When microglia fail to perform this essential function a number of downstream effects can occur and lead to disease development. For example, the failure to keep up with the clearance of Aβ monomers contributes to the formation of Aβ oligomers and eventually plaques, which are a hallmark of AD pathophysiology [73,74,75]. 

One of the metabolic pathways, in addition to cholesterol efflux and storage, that can be activated in the presence of cholesterol is the generation of a variety of cholesterol metabolites, which in turn can either act as regulators of downstream pathways or have direct deleterious effects. Oxysterols are generated in animals, including humans, by enzymatic means as well as non-enzymatic means. Cholesterol can be oxidized to 25-hydroxycholesterol by cholesterol 25-hydroxylase, and to 27-hydroxycholesterol by sterol 27-hydroxylase [76]. While 25-hydroxycholesterol can also be generated by non-enzymatic oxidation by reactive oxygen species (ROS), 7-ketocholesterol is generated exclusively through non-enzymatic means, for example via oxidation by ROS [76]. Oxidized cholesterol is fundamental for generating a pro-inflammatory environment for microglia [77]. In rodent microglia, cholesterol oxides confer cytotoxic effects by potentiating the effects of LPS and nitric oxide production, promoting programmed cell death [78]. Specifically, 25-hydroxycholesterol was observed to induce the highest mRNA levels of nitric oxide synthase in combination with LPS in these cells [78]. Similarly, 27-hydroxycholesterol in vitro treatment of rodent microglia cell lines induced accumulation of ROS and the subsequent activation of the pro-inflammatory interleukin-6/signal transducer and activator of transcription 3 signaling pathway [79]. In turn, in the presence of increased ROS the proportion of sterols of non-enzymatic origin increases, and promotes a chronic DAM state [80]. Studies have also shown that 25-hydroxycholesterol can increase the area of the lipid bilayer as well as affecting the orientation of lipids within the membrane [81], increasing membrane permeability [82,83,84], which has a direct influence on cell death activation [85]. It has also been reported that 27-hydroxycholesterol can induce cellular senescence in microglia through oxidative damage [79,86,87], and that 7-ketocholesterol promotes cellular death by altering biogenesis and peroxisomal activity through oxidative stress [88,89]. Investigators report 7-ketocholesterol released during chronic inflammation indirectly induces neuronal damage mediated by activated microglial cells [90]. It is not yet clear how the relative and absolute concentrations of all of these cholesterol species impact overall microglial function and phenotype.

There is evidence that when microglia are unable to maintain proper cholesterol metabolism and lipid droplets accumulate a pro-inflammatory lipidomic profile emerges [91]. A number of approaches to reduce this cholesterol-induced microglia dysfunction have been investigated. The liver X receptor, which is induced by oxysterols, agonistically promotes an anti-inflammatory environment in the central nervous system (CNS) of rodent models and their primary microglia [92]. Liver X receptor-mediated suppression of inflammation and lipid recycling has also been shown to mitigate disease severity at the microglial level in rodent models [93]. These reports in rodent models suggest that reducing cholesterol via liver X receptor activation could be an approach for clearing the burden from microglia and restoring their functionality.

An additional way in which cholesterol can negatively impact microglia function is through mechanisms involving membrane proteins [94]. The enrichment of cholesterol in plasma membranes potentiates the formation of lipid rafts, which increases the physical proximity of raft-associated proteins. An example of how this can be detrimental when excessive is the case of overactivation via LPS. In a high-cholesterol membrane environment, monomers of Toll-like receptors are in close proximity to each other, enabling the formation of Toll-like receptor dimers, which in turn leads to pro-inflammatory signaling in response to activation by LPS [95]. In murine models chronic LPS activation leads to increased Aβ deposition [96,97]. 

A high cholesterol diet (3% cholesterol) has been shown to induce a pro-inflammatory profile in rodent microglia models by activating the inflammasome; anti-inflammatory cytokines are also secreted as part of the response, but the result is ultimately a damaged blood-brain-barrier [98]. Furthermore, the literature indicates that cholesterol load causes chronic inflammation in microglia [99]. A high fat, high cholesterol diet (21% fat), administered for 18 weeks increased the presence of interleukin-6 in the microglia and plasma of wild-type and APOE-/-mice [100]. The APOE4 isoform of APOE has been consistently associated with an increased risk for AD in genome-wide association studies [101]. The ApoE4 protein encoded by the APOE4 gene has been shown to have a significantly reduced capacity to induce cholesterol efflux from a variety of cell types compared to APOE3 [102]. In a human microglia cell line expressing APOE4 it was observed that excess cholesterol leads to higher levels of inflammation [103], highlighting that a reduced capacity to efflux cholesterol, particularly in an environment of excess cholesterol, is associated with microglial activation. Together, these findings suggest that a high cholesterol environment, particularly in genetically susceptible individuals with a reduced capacity to transport and efflux cholesterol (e.g., APOE4 carriers) leads to chronic microglia inflammation and activation, reducing the ability of microglia to respond to additional stressors.

In the CNS, increasing cholesterol leads to reduced phagocytosis by phagocytes [104], and conversely, depleting cholesterol with methyl-β-cyclodextrin increases phagocytosis. Depleting cholesterol using methyl-β-cyclodextrin enhanced phagocytic activity in primary rat microglia when treated with cholesterol and LPS [105]. Alternatively, one group reported that the accumulation of esterified cholesterol in microglia as a result of the dysfunction of the transmembrane structure triggering receptor expressed on myeloid cells 2, a receptor for lipidated ApoE and other lipids [106,107], did not evoke changes in their phagocytic capacities [108]; suggesting that the concern should not lie solely on the amount of cholesterol microglia are exposed to, but their capacity to traffic cholesterol accordingly. Moreover, microglia cultured from ABCA1 −B/−B mice, exhibit augmented LPS-induced secretion of tumor necrosis factor α (TNF-α) and decreased phagocytic activity hand in hand with decreased ABCA1/APOE expression, which are involved in cellular cholesterol efflux [56,109,110]. Loss-of-function of ABCA7, which also impairs the ability of microglia to efflux cholesterol, accelerated enzymatic activity on the amyloid precursor protein, impaired microglial Aβ clearance and impaired the ability of microglia to perform phagocytosis, contributing to the development of AD [111,112]. In a mouse AD model using a knockout of the protein translocator protein 18 kDa, a molecular sensor specific to glial cells in the brain, it was shown that there is increased Aβ deposition in the brain and a decreased number of microglia undergoing phagocytosis compared to control mice [113], highlighting the importance of effective microglial phagocytosis in the prevention of AD.

Accumulating evidence is demonstrating that defects in cholesterol metabolism, particularly in the context of a variety of high cholesterol environments, are deleterious to microglia function. However, it is not yet clear how to reverse this defective phenotype and correct the underlying cholesterol transport defect in order to improve microglia function. It is also not yet clear whether the DAM phenotype can be reversed.

## 4. Effects of Statins on Microglia

Statins are among the most highly prescribed drugs for the management of hypercholesterolemia (plasma total cholesterol >200 mg/dL) and cardiovascular disease risk. As a result, millions of patients have been prescribed this class of drugs, which inhibit cholesterol biosynthesis. Clinical studies have shown that patients treated with statins, or a combination of statins plus medications to control blood pressure, have a reduced risk for developing AD and dementia [114,115]. Furthermore, patients who were exposed to angiotensin-receptor blockers had lower Aβ deposition in the brain measured with positron emission tomography [116,117]. Statins have been found to have generally anti-inflammatory properties [118], which may be part of their mechanism of action in the context of AD. It has been established that statins have the ability to cross the blood–brain barrier [119]. Thus, it is plausible that this class of drugs could have direct effects on a variety of cells within the CNS, including microglia. In experimental AD rodent models, the statin atorvastatin had a positive impact on cognitive performance and dampened the activated state of microglia, downregulating inflammatory signaling at the mRNA level [120], suggesting that the inhibition of cholesterol synthesis in the context of AD may have beneficial effects via the reduction in neuroinflammation associated with microglia. It has been observed that in microglia statins can lead to a number of effects, including effects on phagocytic activity, on cell activation, and inflammatory signaling.

Firstly, statins have been found to improve phagocytic activity in human microglia and comparable cell models; however, with some unresolved questions about the underlying mechanisms involved [121,122,123,124]. Statins increase phagocytic activity in microglia and glia-like cells in part by inhibiting the secretion of pro-inflammatory signals, including TNF-α [125,126]. However, some studies found that changes in phagocytic activity may be linked to both cholesterol-dependent and cholesterol-independent mechanisms, with both increases and decreases in pro-inflammatory cytokine expression reported [105,127]. Further research is needed to understand both the cholesterol-dependent and cholesterol-independent effects of different statin formulations on microglia phagocytic activity. 

Chronic microglia activation and concurrent pro-inflammatory signaling can result in damage to neighboring cells [90] and in turn diminish the capacity of microglia to undergo normal function [96,97]. The impact of statins on inflammatory signaling has been explored in rat cellular models, where an induced LPS inflammatory response in microglia was dampened by reducing the expression of pro-inflammatory cytokines that include TNF-α [128]. Statins are directly involved in attenuating the effects of TNF-α, and the release of associated factors and receptors in this signaling cascade [120,129]. Furthermore statins can also have an inhibitory effect on the production of superoxide free radical [123,130], protecting cells from oxidative damage, inducing the production of ionized calcium-binding adaptor molecule 1, regulating lipid metabolism and phagocytosis [131,132,133]. 

There are also several lines of investigation suggesting that statins can reduce microglial activation and as a result reduce the propensity for neurodegenerative disease [124,134]. Statins can inhibit microglial activation by blocking the activation of pro-inflammatory cascade pathways [25,130,135] and by promoting the enzymatic degradation of Aβ [136]. Statins have also been found to attenuate the expression of matrixins in microglia, reducing pro-inflammatory signaling [130,137,138], inhibiting the expression of prostaglandin-endoperoxide synthase 2, and redirecting microglia away form an activated state and toward a homeostatic state [130,139]. Studies in human cell lines and in rat neonatal cortical microglia found that atorvastatin reduced the secretion of pro-inflammatory cytokine interleukin-6 [125]. However, simvastatin was not able to reduce pro-inflammatory cytokine interleukin-6 secretion and hindered cell viability in both human microglia cell lines and rat neonatal cortical microglia [125]. Thus, it is important to note that the effects of different statin drugs may be differential with respect to their specific effects on microglia. 

An increase in APOE expression has been demonstrated to be a defining factor in the conversion of homeostatic microglia to DAM [36]. Several lines of evidence have now shown that statins may be important in regulating microglia function in part through their effects on APOE. Administration of statins reduced brain APOE mRNA levels, ApoE protein content and ApoE secretion, attenuating the effects of the high-cholesterol diet [140]. In vitro experiments have also shown that APOE expression can be increased in the CNS in response to cellular stress, and that simvastatin was able to reduce that expression [141]. In mice, inhibition of TNF-α signaling increased APOE mRNA and protein levels, whereas inflammatory signaling otherwise dampens APOE expression in microglia [142]. It is still unclear how these various anti-inflammatory mechanisms are activated by the different statin formulations, and the different hypotheses from different reports dependent on the model under observation [143]. However, the evidence thus far, as shown in Table 1, suggests that statins could be a viable approach, whether directly through cholesterol-mediated or indirectly through cholesterol-independent mechanisms, to modulate microglial activation and functionality. 

## 5. Conclusions

Exposure to excess cholesterol has been shown to drive several aspects of pathological microglia states, including increased inflammatory signaling and decreased phagocytic capacity, both of which have been implicated in AD pathophysiology. The overall picture that is emerging, as depicted in Figure 1, is that in a high cholesterol environment, high cholesterol concentrations in neuronal plasma membranes lead to higher production of Aβ due to colocalization of amyloid precursor protein and γ secretase, which in turn leads to a higher burden of Aβ that needs to be cleared by microglia. At the same time, high cholesterol concentrations impair the ability of microglia to clear Aβ, and increase microglial inflammatory signaling and ROS production. All of this further drives the accumulation of Aβ oligomers and eventually plaque formation, as well as creating a pro-inflammatory environment that contributes to neurodegeneration. The highly prescribed, generally safe, statin drugs may be a viable approach for regulating cholesterol and reducing neuroinflammation but further research on their specific effects on microglia are needed. Further research is also needed to understand how defective or suboptimal cholesterol metabolism impacts microglia phenotype and function so that novel targeted therapies to restore microglia to their functional, homeostatic state can be developed. 

## Figures and Tables

**Figure 1 biomedicines-10-03105-f001:**
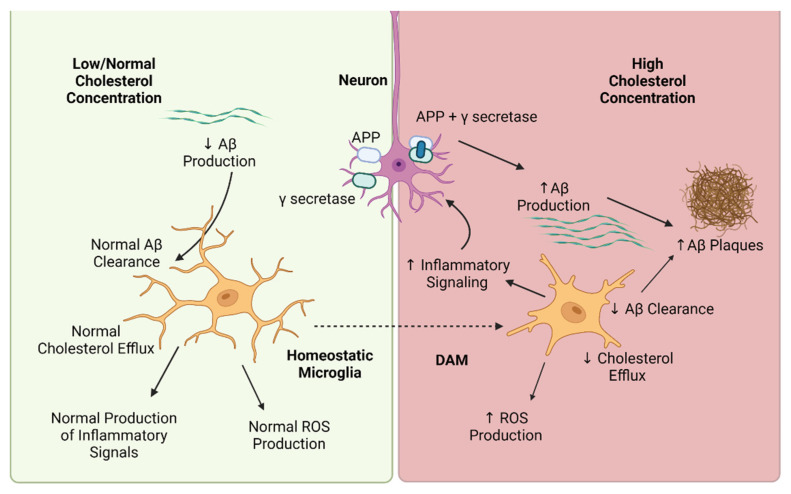
High cholesterol concentrations and disease-associated microglia (DAM). When cholesterol concentrations are low or normal the amount of amyloid-β (Aβ) production is low, coupled with normal microglia Aβ clearance, and a homeostatic microglia phenotype with normal levels of cholesterol efflux, normal production of inflammatory signals and reactive oxygen species (ROS). Conversely, exposure to high cholesterol concentrations leads to increased production of Aβ by neurons due to colocalization of amyloid precursor protein (APP) and gamma secretase in the plasma membrane. Concurrently, high cholesterol concentrations lead to induction of the DAM phenotype, in which increased inflammatory signaling, increased production of ROS, and decreased cholesterol efflux hinder the ability of microglia to clear Aβ, further increasing the concentration of Aβ oligomers and driving plaque formation. Created with BioRender.com.

**Table 1 biomedicines-10-03105-t001:** Summary of studies highlighting the effects of statins on microglia.

Study	Study Type	Model	Statin (Dose)	Outcome Measure	Results
Wang, 2018 [120]	In vivo	Sprague Dawley male rats (age 7 to 8 weeks), 250–300 g	Atorvastatin 5- and 10-mg/kg (chronic)	Number of Iba-1-positive microglia	Reduced number of Iba-1 positive microglia.
Ewen, 2013 [122]	In vivo	Sprague–Dawley male rats (12 week of age)	Atorvastatin 2, 5, and 10 mg/kg	TNF-α and IL-10 levels, and infiltration at site of injury	Atorvastatin decreased TNF-α and increased in IL-10 levels, and number of activated microglia.
Lindberg, 2005 [125]	In vitro and in vivo	CHME-3 human cell line; primary rat microglia	Atorvastatin 0.1, 1, 5, and 20 mM or Simvastatin 0.1, 1, 5, and 10 mM; Atorvastatin 1, 5, and 20 μM	Microglial Secretion of IL-6	Atorvastatin reduce IL-6 secretion of stimulated human and rat microglia.
Townsend, 2004 [127]	In vitro and in vivo	BALB/c mice microglia	Lovastatin 10 µM	IL-6, TNF-α and IL-β1 concentrations and phagocytosis activity	Lovastatin reduced IL-6, TNF-α and IL-β1 concentrations and attenuated impaired phagocytosis in primary mouse microglia.
Pahan, 1997 [128]	In vitro	Isolated primary rat microglia from mixed cultures	Lovastatin 10 µM	Nitiric Oxide, TNF-a, IL-1b, and IL-6 concentrations	In LPS stimulated primary rat microglia, lovastatin reduced nitric oxide, TNF- α, IL-1β, and IL-6 in supernatant.
Yongjun, 2013 [130]	In vitro	Primary human microglia	Atorvastatin 0.1 mM	MT1-MMP expression	Reduced microglia expression of MT1-MMP.
Kata, 2016 [131]	In vitro	Primary rat microglia	Rosuvastatin 1 µM	Iba-1 immunoreactivity; phagocytosis activity; IL-10, IL-1b and TNF-α production.	In microglia challenged with LPS, rosuvastatin reduced IL-1 β, TNF-α production and phagocytosis, IL-10 and Iba1 immunoreactivity was increased.
Chu, 2015 [135]	In vivo	Sprague–Dawley male rats	Atorvastatin 10 mg/kg/day	pNFκB immunostaining	Proinflammatory pNFκB proteins were decreased by atorvastatin in microglia, following surgery.
Tamboli, 2010 [136]	In vivo	BV-2 mouse microglia	Lovastatin 5 µM	Aβ degration, Westen blot	Lovastatin enhanced the degradation of extracellular Aβ by microglial cells.
Petanceska, 2003 [140]	In vivo	C57/BL6 mice and BV-2 cell line	Lovastatin 5 µM and atorvastatin 5 µM	ApoE Western blot	In mice, lovastatin reduced ApoE secretion. Atorvastatin reduced the levels of both cellular and secreted ApoE.

## Data Availability

Not applicable.

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
