# Peer review of "Microglia and Cholesterol Handling: Implications for Alzheimer’s Disease"

_biomedicines, 2022, doi:10.3390/biomedicines10123105_

Round 1
Reviewer 1 Report
This manuscript is of broad interest to readers in the field of microglial inflammation, cholesterol metabolism, and Alzheimer’s disease treatment. Microglia play an essential role in the pathophysiology of AD; microglia can clear Aβ. In this review, the author summarized how cholesterol regulates microglia phenotype and function and how statins affect microglia, the most prescribed drugs to control hypercholesterolemia. The author mentioned that high cholesterol would promote regular microglia change to disease-associated microglia, impairs the ability of microglia to clear Aβ and affects cholesterol efflux, and increase microglia inflammatory signaling; these effects will drive Aβ plaque formation. The references well support the fundamental claim of this manuscript, and this manuscript is directly related to, and support, previous known findings.
Author Response
We thank the reviewer for their time in reviewing the manuscript and for their positive feedback.
Reviewer 2 Report
The paper "Microglia and Cholesterol Handling: Implications for Alzheimer's Disease" is interesting and well-written. Yet, it could be further improved.
1. The authors continuously use the term "high cholesterol"; "hypercholesterolemia." This needs to be clarified. Most likely, different studies covered by this review used different approaches to this term. Therefore, it is best clarified.
2. The last subsection, "Effects of statins on microglia," could benefit from systematization. For this, the authors should consider introducing a table that includes all studies covered. The table should summarize the main findings and indicate the study type, i.e., in silico, pre-clinical, etc.
Author Response
The paper "Microglia and Cholesterol Handling: Implications for Alzheimer's Disease" is interesting and well-written. Yet, it could be further improved.
Point 1. The authors continuously use the term "high cholesterol"; "hypercholesterolemia." This needs to be clarified. Most likely, different studies covered by this review used different approaches to this term. Therefore, it is best clarified.
Response 1. We appreciate the chance to provide more clarity. The information has been expanded on Lines 38-42, 187, 191-192
Point 2. The last subsection, "Effects of statins on microglia," could benefit from systematization. For this, the authors should consider introducing a table that includes all studies covered. The table should summarize the main findings and indicate the study type, i.e., in silico, pre-clinical, etc.
Response 2. We thank the reviewer for this suggestion and opportunity to improve the manuscript. We have created Table 1, introduced in line 301, which summarizes the evidence on the effects of statins on microglia.